# Meta-Learning for Speeding Up Large Model Inference in Decentralized Environments

**Yipeng Du**[*], **Zihao Wang**[*†], **Ahmad Farhan, Claudio Angione, Harry Yang**[†],
**Fielding Johnston, James P. Buban, Patrick Colangelo, Yue Zhao, Yuzhe Yang**
Nesa Research
[†]zwangmn@connect.ust.hk, [†]yangharry@ust.hk, research@nesa.ai

## Abstract

The deployment of large-scale models, such as large language models (LLMs), incurs substantial costs due to their computational demands. To mitigate these costs and address challenges related to scalability and data security, there is a growing shift towards decentralized systems for model deployment, where choosing efficient inference acceleration schemes become crucial to manage computational resources effectively and enhance system responsiveness. In this work, we address the challenge of selecting optimal acceleration methods in decentralized systems by introducing a meta-learning-based framework. This framework automates the selection process by learning from historical performance data of various acceleration techniques across different tasks. Unlike traditional methods that rely on random selection or expert intuition, our approach systematically identifies the best acceleration strategies based on the specific characteristics of each task. We demonstrate that our meta-learning framework not only streamlines the decision-making process but also consistently outperforms conventional methods in terms of efficiency and performance. Our results highlight the potential of inference acceleration in decentralized AI systems, offering a path towards more democratic and economically feasible artificial intelligence solutions.

## 1 Introduction

The growing demand for large-scale models such as language models and image generation systems has significantly increased computational requirements. Although centralized systems remain capable, they often face challenges related to scalability, data security, and cost. These limitations have sparked interest in decentralized architectures, which distribute computation across multiple nodes to improve efficiency, reduce latency, and enhance privacy. (Brown et al., 2020; Ramesh et al., 2022; Zhang et al., 2025). Traditional centralized systems, though powerful, face key limitations in scalability, data security, and cost (Li et al., 2022; Dong et al., 2025). These constraints have driven interest in decentralized architectures that distribute computation across nodes to enhance efficiency, reduce latency, and protect data privacy (Belotti et al., 2019; Zeng et al., 2025; Zhang et al., 2024).

Deploying AI in distributed settings introduces distinct challenges from centralized systems, due to heterogeneous hardware, variable network latency, and dynamic load balancing requirements (Borzunov et al., 2024; Biran & Kissos, 2025; Balseiro et al., 2025). Unlike homogeneous data centers, decentralized systems span diverse devices with varying compute, memory, and communication capabilities. This heterogeneity complicates workload distribution and calls for adaptive strategies responsive to hardware availability and runtime constraints(Shen et al., 2025).

Motivated by these challenges, we evaluate fast inference techniques, such as continuous batching, prefix caching, and chunked prefill—under varying batch sizes and hardware

---

[*]Equal contribution.

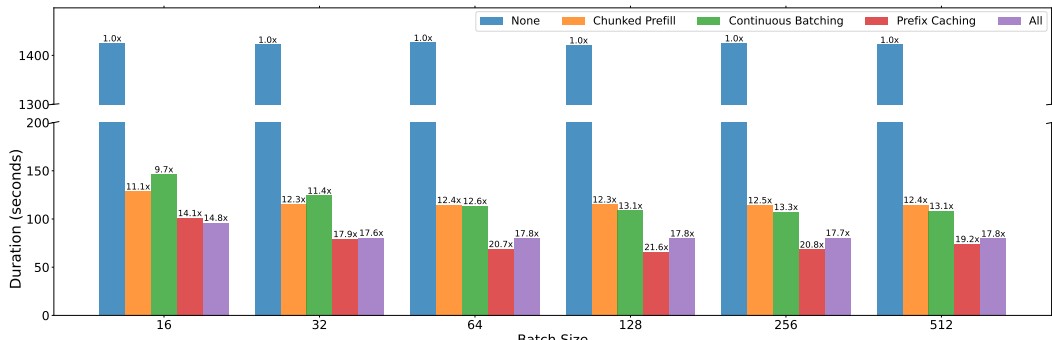

Figure 1: **Performance comparison of acceleration strategies across varying batch sizes.** We compare the performance of different fast inference techniques on the LLaMA 3.1 8B model. The combined strategy outperforms all individual techniques for small batch sizes, while prefix caching performs best for larger batches. This highlights the need for adaptive selection mechanisms tailored to workload scale.

settings. While effective in uniform, high-throughput settings, their utility in heterogeneous systems is less predictable. For instance, prefix caching may underperform on memory-constrained nodes, and chunked prefill can incur communication overhead in low-bandwidth environments. Performance is further modulated by interactions between model architecture, input characteristics, and system state. As shown in Figure 1, combining all techniques excels in small-batch regimes, whereas individual strategies fare better with large batches. This reinforces a central insight: *no single inference strategy is universally optimal*. We further demonstrate this in 1 where Phi-2 performs substantially worse with prefix-caching turned on while other models like Baichuan2 performs significantly better. This demonstrates the need for a predictor that understands the different attributes and can thus pick the correct method for each model and particular hardware.

Hence, achieving efficient inference in decentralized systems requires adaptive scheduling that reasons over models, inputs, and hardware in real time.

| Model | Chunked Prefill | Prefix Caching | All Methods |
|---|---|---|---|
| Baichuan2-7B-Chat | +3.82% | +37.63% | +7.96% |
| Qwen2.5-7B-Instruct-1M | **+4.15%** | −10.66% | −7.20% |
| Phi-2 | −0.90% | −56.79% | −11.39% |
| Meta-Llama-3.1-8B-Instruct | **+5.40%** | −4.03% | +0.33% |

Table 1: Additional experiments on NVIDIA L4 GPUs, with continuous batching turned on by default with fixed batch size of 1024

To this end, we introduce a *learnable meta-scheduler* that predicts optimal inference strategies and hardware configurations from real-time input and system signals. We cast inference optimization as a meta-learning problem, where rich embeddings of workloads and hardware states guide deployment decisions. This flexible framework, detailed in Section 3, addresses the need for robust inference across diverse conditions.

**Our contributions are as follows:**

- We introduce a principled formulation of inference strategy selection as a *budget-constrained meta-learning problem*, where acceleration methods are selected based on learned mappings from task, model, and hardware representations to performance outcomes. This framing enables cost-aware deployment of large models under heterogeneous and decentralized constraints.
- We propose *MetaInf*, a lightweight meta-scheduling framework that leverages LLM-derived semantic embeddings and historical inference metadata to predict optimal

acceleration strategies under deployment constraints. By decoupling selection from online measurement, MetaInf enables zero-shot generalization to unseen model–hardware–workload combinations and maintains high accuracy even under strict cost budgets.

- We develop a unified embedding architecture that encodes heterogeneous configuration spaces—comprising model architecture, hardware profiles, and task descriptions—using language model-derived representations. These embeddings capture latent compatibility across system components and support robust generalization under distribution shift.

## 2 Related Works

**Inference for Large Systems.** Fast inference is essential for deploying large-scale models under constraints of real-time performance, responsiveness, and cost-efficiency (Brown et al., 2020; Rajbhandari et al., 2020). While much prior work targets acceleration in *centralized*, high-performance environments, an emerging body of research addresses the distinct challenges of *distributed* and *heterogeneous* systems.

**Inference in Centralized Systems.** Key techniques have been developed to accelerate inference in centralized settings. **Large batch sizes** improve GPU utilization via request aggregation (Shoeybi et al., 2019; Henighan et al., 2020). **Continuous batching** interleaves requests to reduce idle compute and boost throughput (Yu et al., 2022; Kwon et al., 2023). **Prefilling** and **prefix caching** exploit autoregressive structure to reuse cached key-value pairs, avoiding redundant computation (Pope et al., 2023; Rajbhandari et al., 2020). **Prompt caching** generalizes this reuse by detecting shared prefixes across requests (Brown et al., 2020). Recent advances such as **speculative decoding** employ lightweight draft models to propose candidate outputs, later verified by larger models. This technique accelerates generation while preserving output quality (Leviathan et al., 2023; Cai et al., 2024; Li et al., 2024), and has been widely adopted in fixed, high-throughput inference pipelines.

**Inference in Distributed and Heterogeneous Systems.** While these techniques yield gains in homogeneous clusters, *distributed* and *heterogeneous* environments introduce new challenges—such as diverse compute capabilities, memory limits, and variable network latency (Borzunov et al., 2024; Xu et al., 2019). In edge and decentralized deployments, assumptions valid in centralized systems often break down due to dynamic conditions. Several works address these challenges via expert routing (Lepikhin et al., 2020), model parallelism, and adaptive scheduling (Ren et al., 2021). However, many rely on fixed routing strategies or assume stable infrastructure, limiting adaptability in real-world, non-stationary settings.

**Towards Adaptive Distributed Inference.** Our work frames inference optimization as a dynamic decision-making problem. Instead of static heuristics, we propose a learnable meta-scheduler that adapts in real time to hardware profiles, input complexity, and system conditions. This approach bridges fine-grained acceleration with system-level adaptability, enabling scalable, robust AI deployment.

## 3 Method

### 3.1 Motivating Example: Performance Variability in Inference Acceleration

To motivate the need for a meta-learning-based scheduler, we first conduct a comprehensive evaluation of popular inference acceleration techniques across heterogeneous hardware and model configurations. Our goal is to highlight the variability and unpredictability of method performance, thereby justifying the need for adaptive selection strategies.

| Method | BS=16 | BS=256 |
|---|---|---|
| None | 1435.27 | 1424.99 |
| CP (Chunked Prefill) | 128.70 | 114.44 |
| CB (Cont. Batch) | 146.44 | 107.40 |
| PC (Prefix Cache) | 101.10 | **68.46** |
| All | **96.21** | 80.65 |

Table 2: **Latency vs. batch size.** Inference latency (s) for each strategy at BS 16 and 256. PC excels at large BS, while combining all works best at small BS.

| Method | 4 GPUs | 8 GPUs |
|---|---|---|
| CP | 120.89 (5.43) | 117.21 (5.15) |
| CB | 122.73 (12.75) | 118.19 (13.87) |
| PC | 79.18 (11.86) | 76.33 (11.92) |
| All | 85.07 (6.04) | 82.90 (5.96) |

Table 3: **Scaling with GPUs.** Runtime (s, var.) for 4 vs. 8 GPUs. Diminishing returns appear for CP and CB due to comm. overhead.

### 3.1.1 Evaluation Setup

We implemented continuous batching, prefix caching, and chunked prefill, as well as their combined use, within a unified inference engine. All experiments were conducted using the ShareGPT dataset (Kwon et al., 2023).

**Models.** We evaluated widely used open-source instruction-tuned LLMs, primarily Meta-Llama-3.1-8B-Instruct, since many open-source variants adopt similar architectural patterns.

**Hardware.** Inference was executed on NVIDIA L4 GPUs using tensor parallelism across 4 and 8 GPU configurations.

**Configuration.** All models were capped at a maximum output length of 2048 tokens. Quantization precision was maintained at float16.

### 3.2 Findings and Insights

- **Inference acceleration methods substantially improve throughput.** Continuous batching and prefix caching yielded up to **13×** and **20×** speedups respectively compared to unoptimized baselines (Table 2).
- **No method is universally optimal.** As shown in Figure 1, prefix caching performed best for large batches, while continuous batching excelled in low-latency settings. The "All" configuration performed well in some setups but introduced overhead in others.
- **Hardware scaling shows diminishing returns.** Scaling from 4 to 8 GPUs led to only modest throughput gains (~5%), suggesting that communication overhead and bandwidth constraints limit performance.

These observations reveal that acceleration performance depends not only on the method itself but also on contextual factors such as model architecture, batch size, and hardware topology. This motivates our proposed solution: a learnable meta-scheduler that can adaptively select acceleration strategies given a target system state.

### 3.3 Problem Statement and Framework Overview

Given a new model, dataset, and hardware environment for distributed inference, our goal is to select the best acceleration method—such as quantization, model compression, or parallelism—*without* costly empirical evaluations. This selection must account for hardware-specific constraints and computational capabilities.

We leverage meta-learning to transfer performance knowledge from prior inference tasks. The key insight is that methods effective in similar historical contexts are likely to perform well on new configurations, especially when immediate evaluations are infeasible due to deployment urgency or resource constraints.

Our proposed meta-learner, MetaInf, optimizes inference acceleration across diverse environments by learning from historical data. It comprises the following components:

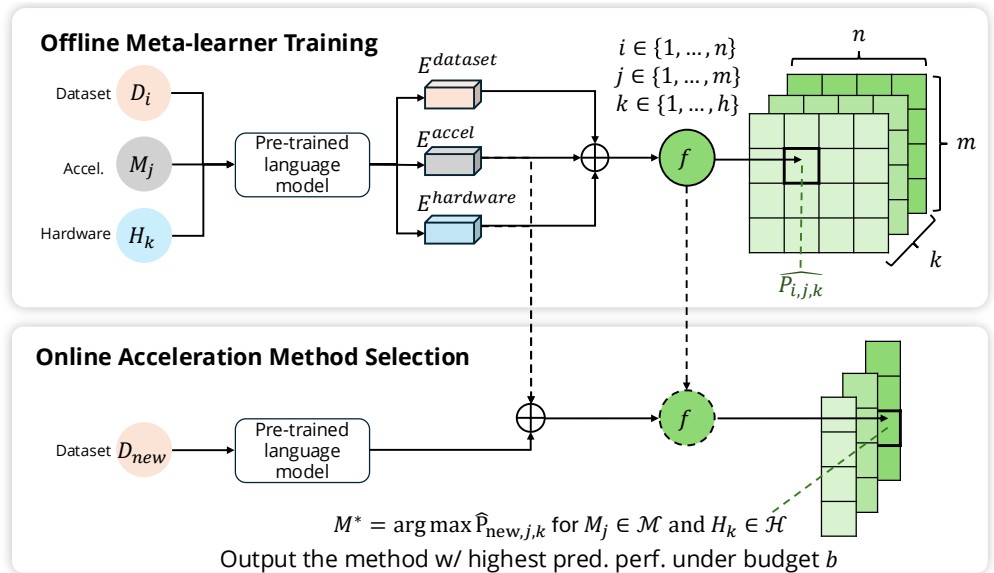

Figure 2: **MetaInf overview** (§3.3). The top illustrates offline meta-training (§3.4), where a predictor $f$ (shown as ●) learns to map embeddings of datasets and models to performance outcomes **P**. The bottom shows online selection (§3.5), where $f$ is used to predict performance for new tasks and hardware settings.

- A database of historical inference tasks, $\mathcal{D}_{\text{train}} = \{D_1, \ldots, D_n\}$, defines a broad experience base. Each task corresponds to a distinct model input, e.g., prompts for LLMs.
- A catalog of hardware configurations, $\mathcal{H} = \{H_1, \ldots, H_h\}$, captures how different platforms influence method effectiveness and cost.
- A performance matrix $\mathbf{P} \in \mathbb{R}^{n \times m \times h}$ records the metrics of predefined methods $\mathcal{M} = \{M_1, \ldots, M_m\}$ across tasks and hardware. Each entry $\mathbf{P}_{i,j,k}$ quantifies the performance of method $M_j$ on task $D_i$ using hardware $H_k$.

We aim to select an acceleration method $M \in \mathcal{M}$ for a new task that maximizes performance while satisfying a deployment budget $b$. The deployment cost on hardware $H_k$ is defined as the product of runtime and hardware cost, which must not exceed $b$.

**Problem 1 (Cost-Constrained Acceleration Method Selection)** *Given a new task $D_{new} = \{\mathbf{X}_{train}^{new}, \mathbf{X}_{test}^{new}, \mathbf{H}_{new}\}$ and a cost budget $b$, select an acceleration method $M \in \mathcal{M}$ such that the cost of using $M$ on hardware $H_k$ for $D_{new}$ is minimized while not exceeding $b$.*

MetaInf operates in two phases: (*i*) offline meta-training on $\mathcal{D}_{\text{train}}$ to learn performance mappings, and (*ii*) online method selection, where these mappings guide efficient deployment of new tasks. Figure 2 summarizes this workflow, with full details in Sections 3.4 and 3.5.

### 3.4 Offline Meta-Training

In the offline phase, we generate embeddings that encode dataset characteristics, acceleration methods, and hardware environments. These representations are used to learn a mapping to performance metrics **P**.

$$f : E_i^{\text{data}}, E_j^{\text{model}}, E_k^{\text{hardware}} \to \mathbf{P}_{i,j,k}, \tag{1}$$
$$i \in \{1, \ldots, n\}, \quad j \in \{1, \ldots, m\}, \quad k \in \{1, \ldots, h\}$$

Embeddings $E^{\text{data}}$, $E^{\text{model}}$, and $E^{\text{hardware}}$ represent factors influencing the effectiveness of acceleration methods under resource constraints. A regression-based meta-predictor $f$

learns to map these embeddings to predicted performance. While one-hot encodings are considered in ablation, we primarily use LLM-derived embeddings, reduced via truncated Single Value Decomposition (SVD), to capture semantic relationships efficiently across tasks and systems.

**Dataset Embedding** ($E^{\text{data}}$): Encodes data volume, distribution, and complexity—key factors in method suitability.

**Method Embedding** ($E^{\text{model}}$): Captures compute/energy cost, accuracy trade-offs, and method-specific traits (e.g., prefix caching).

**Hardware Embedding** ($E^{\text{hardware}}$): Encodes processor type, memory, and power constraints, all critical for feasibility.

We train $f$ using historical records spanning datasets, methods, and hardware settings. For this, we adopt XGBoost (Chen & Guestrin, 2016) due to its robustness to heterogeneous, high-dimensional inputs and interpretability via feature importance.

**Objective**: The goal is to learn a reliable function $f$ that maps ($E^{\text{data}}, E^{\text{model}}, E^{\text{hardware}}$) to **P**—which may include runtime, energy, or accuracy—enabling accurate prediction of the best method for a new task $D_{\text{new}}$.

The complete pipeline is shown in Figure 2 and formalized in Algorithm 1.

### 3.5 Online Model Selection

Given a new task $D_{\text{new}}$, we compute embeddings for its dataset, the available acceleration methods, and the target hardware environment. These embeddings are input to the pretrained meta-performance predictor $f$, which estimates the performance of each acceleration method under current constraints. The optimal method $M^*$ is selected to maximize predicted performance while satisfying a cost budget $b$.

$$M^* := \underset{M_j \in \mathcal{M}, C_{j,k} \leq b}{\arg\max} \widehat{\mathbf{P}}_{\text{new},j,k},$$
$$\widehat{\mathbf{P}}_{\text{new},j,k} = f(E^{\text{data}}_{\text{new}}, E^{\text{model}}_j, E^{\text{hardware}}_k) \tag{2}$$

$C_{j,k}$ represents the cost of running method $M_j$ on hardware $H_k$ and must not exceed the budget $b$. It is calculated as the product of the cost associated with $H_k$ and the expected runtime of $M_j$ on $H_k$.

Thus, for each new task defined by $D_{\text{new}} = \{\mathbf{X}^{\text{new}}_{\text{train}}, \mathbf{X}^{\text{new}}_{\text{test}}, \mathbf{H}_{\text{new}}\}$, embeddings for the dataset ($E^{\text{data}}_{\text{new}}$), the applicable model ($E^{\text{model}}_j$), and the hardware environment ($E^{\text{hardware}}_k$) are computed. The trained meta-predictor $f$ is then used to evaluate the performance of each available acceleration method under the specific hardware settings of $\mathbf{H}_{\text{new}}$, ensuring that the total cost remains within the budget.

These embeddings are derived from LLM-based textual representations and reduced using SVD (see §4.2). No retraining or empirical evaluation is required at inference time—selection is performed in a zero-shot manner using the pretrained meta-learner $f$, enabling rapid, scalable deployment across novel settings.

The algorithm applies a constrained maximization strategy to select the highest-performing method under budget $b$. This balances inference quality, latency, and hardware cost, tailored to the unique properties of each task and deployment environment. Figure 2 (bottom) illustrates this process in the online phase. Algorithm 2 formally describes the selection procedure.

## 4 Experiments

We assess the accuracy and efficiency of our meta-scheduler across multiple LLM architectures, hardware platforms, and real-world workloads. For each test instance, the meta-learner selects an acceleration method conditioned on the model and GPU configuration. We then compare the predicted choice against the empirically optimal configuration, determined via full evaluation of all candidate strategies. Performance metrics include selection accuracy, F1 score, and average acceleration ratio. Full details of the experimental protocol are provided in Appendix A.2.

### 4.1 Experimental Settings

**Model Set.** We evaluate four widely adopted generative LLMs: `Baichuan2-7B-Chat`, `Qwen2.5-7B-Instruct-1M`, `phi-2`, and `Meta-Llama-3.1-8B-Instruct`. These models span diverse architectures and training objectives, enabling us to assess acceleration strategies across heterogeneous LLM backbones.

**Dataset.** We use two representative benchmarks: (1) `reasoning-v1-20m`, a synthetic dataset for multi-domain and symbolic reasoning; and (2) `ShareGPT`, a real-world conversational dataset reflecting broad user interaction patterns. This pairing covers both structured and open-ended inference regimes.

**Hardware Platform.** Experiments span NVIDIA T4 (16GB), L4 (24GB), and A100 (40GB) GPUs, capturing deployment on both edge and datacenter-class hardware.

**Acceleration Methods.** We assess three fast inference techniques:

- **Prefix Caching.** Reuses key-value pairs from input prefixes to avoid redundant decoding computation; ideal for long or multi-turn contexts.
- **Chunked Prefill.** Splits the prefill stage into smaller segments to lower memory overhead and reduce initial latency.
- **Continuous Batching.** Dynamically batches incoming requests to maximize GPU utilization without relying on fixed batch sizes.

**Baselines.** We compare our proposed *MetaInf* scheduler against a diverse set of baselines spanning heuristic algorithm selectors and standard machine learning models. Specifically, we evaluate:

- **Heuristic methods:** ISAC (Kerschke et al., 2018), Global Best, ARGOSMART (Agakov et al., 2004), and ALORS (Lindauer & Hutter, 2015), which have been widely adopted in algorithm configuration and meta-scheduling contexts.
- **Learning-based models:** Ridge Regression (Hoerl & Kennard, 1970), Random Forests (Breiman, 2001), Gradient Boosting (Friedman, 2001), LightGBM (Ke et al., 2017), Support Vector Machines (Cortes & Vapnik, 1995), and Multi-Layer Perceptrons (MLP).

These baselines balance interpretability and model capacity, supporting a comprehensive evaluation of both inference performance and selection accuracy.

### 4.2 Data, Model, and Hardware Embeddings

To enable generalization across heterogeneous settings, we use structured embeddings for datasets, models, and hardware. While we conduct ablation studies using one-hot encodings to assess the impact of embedding choice, our default approach employs LLM-generated embeddings that encode rich semantic priors. These embeddings leverage the pretrained knowledge in large models, providing meaningful representations even in zero-shot configurations. To manage the dimensionality of these dense representations, we apply truncated SVD, preserving salient information while improving computational efficiency.

*Dataset Example:* `"This dataset has 10,000 samples with 20 features for regression. High variability adds fitting difficulty."`

| Method | Acc. | F1 | Accel. |
|---|---|---|---|
| ISAC | 0.578 | 0.60 | 1.10 |
| Global Best | 0.629 | 0.63 | 1.16 |
| ARGOSMART | 0.66 | 0.62 | 1.17 |
| ALORS | 0.725 | 0.71 | 1.20 |
| SVM | 0.802 | 0.79 | 1.28 |
| MLP | 0.783 | 0.75 | 1.24 |
| Random Forest | 0.742 | 0.69 | 1.25 |
| LightGBM | 0.737 | 0.68 | 1.19 |
| Ridge Regression | 0.784 | 0.73 | 1.27 |
| Gradient Boosting | 0.815 | 0.78 | 1.30 |
| **MetaInf (Ours)** | **0.898** | **0.85** | **1.55** |

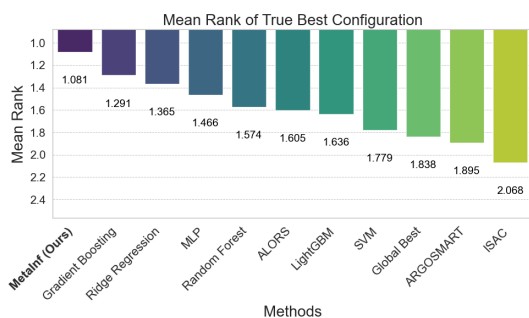

Table 4: **Performance of our MetaInf and baselines methods.** Our MetaInf compared with other prediction methods on selection accuracy (Acc.), F1 score, and acceleration ratio (Accel.) The results show that MetaInf has the best performance compared to other prediction methods.

Figure 3: **Performance of our MetaInf and baseline methods on select the best-performing acceleration combination.** This bar plot shows the average rank of each method when choosing the true best configuration across all tasks. Lower values indicate more reliable and consistent performance; MetaInf (ours) ranks first with the best performance.

*Model Example:* `"Transformer decoder with 8 billion parameters, instruction-tuned for dialogue and instruction following, optimized for inference."`

*Hardware Example:* `"NVIDIA A100 for high-throughput tasks; additional CPU-only machines for lightweight jobs."`

### 4.3 Method Selection Performance

Table 4 presents empirical results comparing our meta-learning-based method, MetaInf, for selecting fast inference techniques against existing approaches. Our method demonstrably outperforms baseline techniques, achieving significantly higher accuracy in selecting the optimal acceleration scheme. The Average Acceleration Ratio, defined here as the ratio of inference time using the prediction model's selected solution to the average inference time across all solutions, quantifies the inference time and cost savings achieved by each method. MetaInf exhibits a superior Average Acceleration Ratio compared to alternative methods, indicating more substantial reductions in inference time and cost. These findings underscore the effectiveness of adaptive selection, as implemented in MetaInf, for accelerating inference in large language models, offering a pathway towards more efficient and scalable deployment.

### 4.4 Performance Trade-off

Figure 5 visualizes the trade-off between prediction accuracy and inference time across different methods. MetaInf emerges as the most effective approach, exhibiting the highest accuracy with a fast inference time. While other methods prioritize either speed or accuracy, MetaInf effectively balances both, demonstrating its practical advantage in achieving high performance with efficiency.

### 4.5 Mean rank of the true best configuration

To evaluate the methods' ability to identify optimal configurations, Figure 3 presents the mean rank of the true best configuration. This metric quantifies how well each method ranks the empirically best configuration amongst its predictions. The process involved, for each model-GPU pair, predicting inference durations for all configurations and ranking them by predicted duration. Then, for each pair, we recorded the rank of the actual best

| Model | GPU | MetaInf Time (s) | Continuous Batching (s) | Time Saved |
|-------|-----|------------------|-------------------------|------------|
| Mistral-7B | A100 | 179.41 | 202.90 | **−11.6%** |
| Mistral-7B | H200 | 31.47 | 38.20 | **−17.6%** |
| Mixtral-8x7B | A100 | 215.86 | 241.92 | **−10.8%** |
| Mixtral-8x7B | H200 | 39.85 | 37.42 | **+6.5%** |
| LLaMA-3.1-70B | A100 | 895.24 | 1012.66 | **−11.6%** |
| LLaMA-3.1-70B | H200 | 150.52 | 168.71 | **−10.8%** |

Table 5: Evaluation of MetaInf in a zero-shot setting on unseen model–GPU pairs. While A100 appears in the training data, H200 does not and was also absent from the embedding model's training corpus. No retrieval-augmented generation (RAG) was used to avoid latency overhead, preserving real-time deployability. The meta-learner selected acceleration strategies which are compared to baseline continuous batching times. MetaInf shows consistent speedups across most settings, even for large dense models like LLaMA-3.1-70B.

configuration within this predicted ranking. Averaging these ranks, where a lower mean rank indicates superior performance in prioritizing truly optimal configurations. The results show MetaInf achieving the lowest mean rank, demonstrating its effective identification of best configurations. These findings coherently underscore MetaInf's enhanced ranking capability, leading to more effective configuration selection for fast inference.

### 4.6 Ablation Study

To evaluate the impact of prompt engineering on the downstream performance of meta-learning-based configuration selection, and whether the model is able to utilize knowledge within the LLM, we conduct an ablation study comparing three levels of prompt complexity as well as different levels of SVD dimensionality reduction:

- **Basic Prompt**: Minimal keyword-based descriptors, e.g., ''`Model: LLaMA-7B`''.
- **Rich Prompt**: Structured templates containing architectural and operational context (e.g., compute intensity, model family, memory constraints).
- **Chain-of-Thought (CoT) Prompt**: Extended, reasoning-driven prompts that encode performance-relevant dependencies and conditional reasoning.
- **SVD Dimensionality**: We vary the number of latent dimensions per embedded feature between $k = 64$, and 256.

Each prompt style is processed by an instruction-tuned LLM (LLaMA-3.2B-Instruct) to produce dense semantic embeddings for five configuration axes: model name, GPU architecture, and three boolean inference flags.

Figure 4 reports the evaluation results averaged over 1000 random (model, GPU) trials. We observe that higher SVD dimensionality improves accuracy and true improvement up to $k = 256$, beyond which performance may plateau.

## 5 Conclusion

In this paper, we introduced a novel meta-learning framework designed to optimize inference acceleration within decentralized systems. By addressing the unique challenges associated with deploying large-scale models such as LLMs, our framework significantly enhances the adaptability and effectiveness of acceleration techniques. Our results demonstrate its superiority over traditional acceleration methods. We also note that by simply switching out our optimization goal, we can turn this framework into assessing more attributes in compute resources other than inference time and cost, such as voltage and power requirements and thus may have a significant impact on our environment as our data centers for AI inference is growing rapidly in today's day and age.

The successful application of our framework highlights its potential to facilitate rapid, efficient inference processes, thereby supporting the broader adoption and democratization

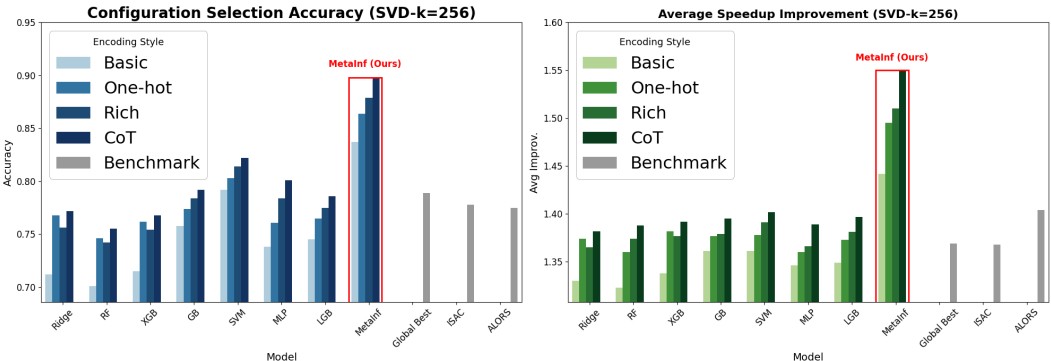

Figure 4: **Performance comparison of MetaInf and baseline methods across different embedding schemes.** We evaluate model selectors under four embedding schemes—One-hot, Basic, Rich, and Chain-of-Thought (CoT)—using configuration selection accuracy (left) and average speedup improvement (right). *MetaInf* consistently outperforms both heuristic and learning-based baselines across all encoding schemes, demonstrating strong generalization and adaptability under embedding variation.

of advanced AI technologies in decentralized settings. Looking forward, this work sets a solid foundation for future research focused on refining and expanding the capabilities of AI systems to meet the increasing demands of real-world applications. We believe that the methodologies developed here will inspire further innovations in the field, particularly in enhancing the operational efficiency of large model infrastructures.

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

# A  Appendix

## A.1  Pseudo-code for Meta-train and Online Model Selection

---

**Algorithm 1** Offline Meta-Learner Training for Inference Acceleration

---

**Input:** Meta-train set $\mathcal{D}_{\text{train}}$, model set $\mathcal{M}$, hardware set $\mathcal{H}$
**Output:** Meta-learner $f$ for acceleration method selection
 1: Train and evaluate $\mathcal{M}$ across $\mathcal{H}$ on $\mathcal{D}_{\text{train}}$ to get performance tensor $\mathbf{P}$
 2: **for** $i \in \{1, \ldots, n\}$ **do**
 3:     Extract data embedding $E_i^{\text{data}} = \psi(D_i)$
 4:     **for** $j \in \{1, \ldots, m\}$ **do**
 5:         Encode method set as $E_j^{\text{model}} = \phi(\mathcal{M}, M_j)$
 6:         **for** $k \in \{1, \ldots, h\}$ **do**
 7:             Encode hardware setup as $E_k^{\text{hardware}} = \theta(H_k)$
 8:             Train $f$ by Eq. (1) using $(E_i^{\text{data}}, E_j^{\text{model}}, E_k^{\text{hardware}})$ to predict $\mathbf{P}_{i,j,k}$
 9:         **end for**
10:     **end for**
11: **end for**
12: **return** the meta-learner $f$

---

---

**Algorithm 2** Online Acceleration Method Selection

---

**Input:** meta-learner $f$, dataset $\mathbf{X}_{\text{new}}$, hardware $\mathbf{H}_{\text{new}}$, budget $b$
**Output:** Selected acceleration method for $\mathbf{X}_{\text{new}}$
 1: $E_{\text{new}}^{\text{data}} := \psi(\mathbf{X}_{\text{new}})$                         ▷ Extract data embedding
 2: $E_{\text{new}}^{\text{hardware}} := \theta(\mathbf{H}_{\text{new}})$               ▷ Extract hardware embedding
 3: **for** $j = 1$ to $m$ **do**
 4:     $E_j^{\text{model}} = \phi(M_j)$                              ▷ Encode method
 5:     $\widehat{\mathbf{P}}_{\text{new},j} := f(E_{\text{new}}^{\text{data}}, E_j^{\text{model}}, E_{\text{new}}^{\text{hardware}})$       ▷ Predict performance
 6:     $C_{j,\text{new}} := \text{cost}(\mathbf{H}_{\text{new}}) \times \text{runtime}(M_j, \mathbf{H}_{\text{new}})$       ▷ Calculate cost
 7:     **if** $C_{j,\text{new}} \leq b$ **then**
 8:         Consider $M_j$
 9:     **else**
10:         Exclude $M_j$
11:     **end if**
12: **end for**
13: $M^* := \underset{M_j \in \mathcal{M} \text{ and } C_{j,\text{new}} \leq b}{\arg\max} \widehat{\mathbf{P}}_{\text{new},j}$
14: **return** $M^*$                ▷ Return method with highest performance within budget

---

## A.2  Additional Experiment Details

The experimental procedure involves using untrained data from the `reasoning-v1-20m` and `ShareGPT` datasets to generate 1,000 random inputs. These inputs are then evaluated using the pretrained prediction methods. Both the reasoning model and GPU platform are randomly selected, while the prediction method identifies the combination of acceleration strategies expected to result in the shortest reasoning time. Following this, the actual reasoning is performed for all acceleration schemes using the previously randomly selected model and GPU configuration, with the same prompt as input. The fastest reasoning configuration, as verified through empirical testing, is used to calculate the accuracy of the acceleration scheme selected by the prediction method.

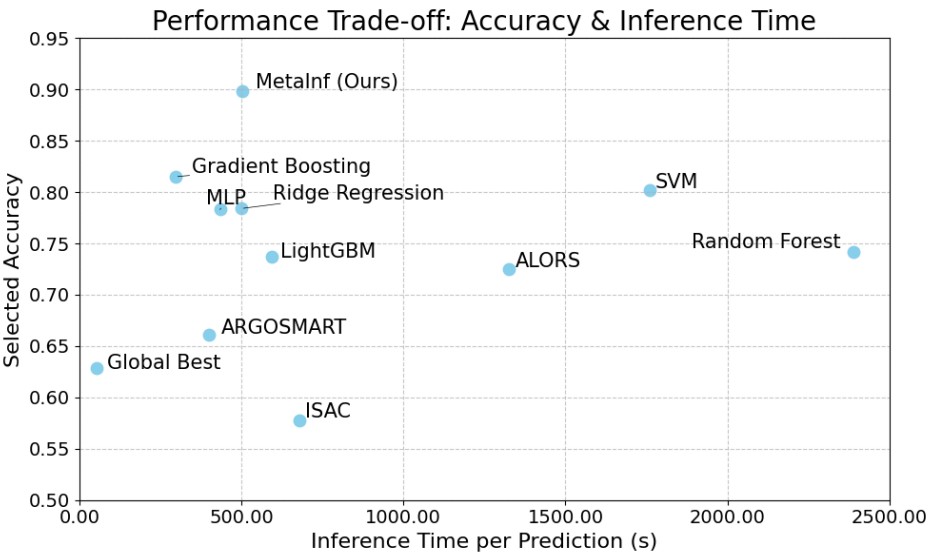

Figure 5: **Efficiency–effectiveness trade-off of prediction methods.** We compare the Efficiency-effectiveness between the accuracy of the tested prediction methods and the inference time of each prediction to measure the performance of our proposed MetaInf compared to other methods. MetaInf shows highest accuracy and lower inference cost.

