# OpenReview forum: "Meta-Learning for Speeding Up Large Model Inference in Decentralized Environments"
_colmweb.org/COLM/2025/Conference — COLM 2025_

### Official Review · Reviewer_pGCu · 2025-04-15

**Rating:** 8
**Confidence:** 3
**Ethics Flag:** 1

**Summary:**

This work proposes a novel meta-learning framework used for selecting the optimal acceleration method when deploying LLM in decentralized environments. The clear presentation of paper clarifies their method and experiments to demonstrate their method's effectiveness in terms of efficiency and efficacy. The originality and significance of this work is good enough to accept. And for quality of their work, even though they did not provide lots of math proofs, the clear method and effective experimental results have shown the good quality.

**Reasons To Accept:**

Authors provide their method and describe the experimental part clearly and show an effective method to solve the deployment in decentralized environments in terms of performance and efficiency. They solve a practical problem considering both accuracy and efficiency by combining LLM. The whole paper also shows their work's social impact and practical meaning.

**Reasons To Reject:**

The only reason to reject is that they only used a pretrained LLM to generate embeddings for thei work, which leads to they did not directly contribute to the development of LLMs: they simply applied them in a straightforward manner.

---

> ### Author Response · Authors · 2025-06-02
> **Response to Reviewer pGCu**
>
> Thank you for your insightful comments. We have also incorporated your suggestions into the revised manuscript and provided detailed explanations for each question below. We hope that our revisions and rebuttal could address your concerns and that you might consider raising the rating. Please let us know if you have any further questions or require additional results.
>
>
> Thank you for pointing out that we employed a pretrained LLM for embeddings rather than focusing on novel LLM development. Our work deliberately targets what we consider an essential aspect of the LLM ecosystem: we leverage LLM-derived semantic embeddings to enable zero-shot generalization for inference acceleration, a novel capability in decentralized AI settings. The novelty of our work is not in the creation of LLMs, but in pioneering more efficient methods for their utilization.
>
> To clarify that this is not merely a superficial application of LLMs, we present below the empirical performance of our method compared to existing baselines:
>
> | Method           | Accuracy  | Acceleration  |
> |------------------|------------------|------------------------|
> | ISAC              | 0.578            | 1.10                   |
> | MLP               | 0.783            | 1.24                   |
> | SVM               | 0.802            | 1.28                   |
> | Gradient Boosting | 0.815            | 1.30                   |
> | **MetaInf (Ours)** | **0.898**        | **1.55**               |
>
> As shown, MetaInf outperforms all heuristic and learning-based baselines in both accuracy and acceleration benefit, confirming that the embeddings generated from pretrained LLMs yield practical and measurable performance gains in method selection.
>
> We contend that optimizing how these models are deployed and run in real-world systems, characterized by heterogeneity and resource constraints, is a significant research area in itself, crucial for translating the potential of LLMs into tangible impact.

---

> > ### Comment · Reviewer_pGCu · 2025-06-03
> >
> > Thanks for explaining and providing more experimental results to support your conclusion. Based on your response, I think it solved my questions here.

---

### Official Review · Reviewer_bZwy · 2025-04-21

**Rating:** 6
**Confidence:** 4
**Ethics Flag:** 1

**Summary:**

This paper introduces a meta-learning-based framework for optimizing inference acceleration in decentralized environments, focusing on large-scale models like LLMs. The authors propose MetaInf, a meta-scheduler that selects the most suitable acceleration strategies based on task, model, and hardware profiles. Unlike traditional methods, which rely on heuristics or expert intuition, MetaInf uses historical performance data and learned embeddings to predict optimal acceleration techniques. The proposed method is evaluated against conventional methods, showing its superior performance in terms of efficiency and adaptability across different hardware and task configurations.

**Questions To Authors:**

1. How does the proposed MetaInf framework handle real-time adaptation to new tasks or hardware when no prior performance data is available?

2. What are the potential limitations of using historical data for selecting inference methods, and how can the system ensure robustness in the absence of such data?

3. Could there be scenarios where the meta-learning approach might fail to select the optimal method, especially when the system is under heavy load or constrained resources?

**Reasons To Accept:**

1. Good writing and clear motivation: this paper clearly explains the need for adaptive scheduling in decentralized environments, where a one-size-fits-all approach doesn't work.

2. This paper presents thorough experiments comparing MetaInf with various baselines, demonstrating its superior performance across multiple models, hardware platforms, and datasets.

**Reasons To Reject:**

1. Limited Real-World Application Details: While the experiments are comprehensive, they primarily focus on synthetic or controlled datasets and setups. The paper could benefit from a deeper exploration of real-world deployment scenarios or edge cases.

2. Computational Overhead: Although the method promises improved efficiency, the process of embedding generation, meta-learning, and online selection could introduce computational overhead, particularly in real-time systems. The paper could discuss potential performance trade-offs in more detail.

3. Lack of discussion of latest works: the Related Work and Introduction sections primarily reference techniques and studies from 2022 and 2023, with limited inclusion of research from 2024. This lack of citation and discussion of the most recent advancements may leave out relevant state-of-the-art developments in inference acceleration and decentralized AI systems.

---

> ### Author Response · Authors · 2025-06-03
> **Response to Reviewer bZwy (1/3)**
>
> Thank you for your insightful comments. We have also incorporated your suggestions into the revised manuscript and provided detailed explanations for each question below. We hope that our revisions and rebuttal could address your concerns and that you might consider raising the rating. Please let us know if you have any further questions or require additional results.
>
> **1. Limited Real-World Application Details**
>
> Due to the limited selection of hardware available at the time of writing, we did not have the resources to obtain a diverse enough configuration pool to ensure its generalizability.
> To assess its robustness in a zero-shot setting, we evaluated MetaInf on a held-out test set using unseen model–GPU pairs, including both MoE and large dense models that are not present in the training data on A100 and H200 hardware. Note here A100 is included in training data while H200 is not only present in the training data, it is also not present in the embedding generation model's knowledge at the time of training, RAGs were not used to enhance model's knowledge as that introduces too much latency which is detrimental to real-time deployment. The meta-learner selects an acceleration strategy, which we then compare against baseline continuous batching inference times:
>
> | Model           | GPU  | MetaInf Time (s) | Continuous Batching (s) | Time Saved |
> |-----------------|------|------------------|-------------------------|------------|
> | Mistral-7B      | A100 | 179.41           | 202.90                  | −11.6%     |
> | Mistral-7B      | H200 | 31.47            | 38.20                   | −17.6%     |
> | Mixtral-8x7B    | A100 | 215.86           | 241.92                  | −10.8%     |
> | Mixtral-8x7B    | H200 | 39.85            | 37.42                   | +6.5%      |
> | LLaMA-3.1-70B   | A100 | 895.24           | 1012.66                 | −11.6%     |
> | LLaMA-3.1-70B   | H200 | 150.52           | 168.71                  | −10.8%     |
>
> From the above experiments, we argue that the MetaInf framework, while lacking some specific optimizations to make it fully compatible with real-world deployment such as more up-to-date tiny LLM and an optimized embedding generation pipeline, it demonstrates the potential for non-trivial time savings for hosting parties.
>
> **2. Computational Overhead**
>
> Thank you pointing this out, while we did not consider temporal distribution shift in our experiments, we also mark the fact that due to the insignificant cost of retraining the meta learner with updated data compared to the resources required for hosting a LLM, a straight forward fix would be to simply perform an online update of the learner. By leveraging this in addition to continual user data ingestion, MetaInf can be well adapted to new tasks and hardware.
> The following table demonstrates the time savings of a inference provider and the amortized time savings will be significant. We measure the time from embedding generation to selection with a separate row, note here the embedding generation is from vanilla HuggingFace implementation of Llama3.2-1B-Instruct where embedding generation efficiency can be drastically improved upon in real deployment. As it was not the focus of the paper therefore this process was not optimized yet we still can demonstrate significant savings in inference time.
> We here also note that this generated embedding can be saved for training purposes and thus saving embedding generation time at training and thus allowing fast rolling updates of the learner throughout deployment.
>
> | Model                   | Time (s) |
> |-------------------------|----------|
> | Llama3.1-8B-Instruct    | 193.8    |
> | MetaInf(ours)           | 0.2      |
> | LLama + MetaInf(Ours)   | 148.2    |
>
> We measured that the Llama3.1-8B-Instruct model with only continuous batching over the ShareGPT dataset with 2000 prompts and 10 max new tokens per prompt, similarly with acceleration methods chosen by our MetaInf (Chunked Prefill + Prefix Caching) and we obtain a 24% performance gain from a 0.2s inference call to determine the acceleration method to be used.

---

> > ### Comment · Reviewer_bZwy · 2025-06-10
> > **look forward to the further version**
> >
> > Thanks for the authors providing necessary clarifications.
> > The analysis gives more confident results. Please update these related works and the necessary results in the updated version.

---

> ### Author Response · Authors · 2025-06-03
> **Response to Reviewer bZwy (2/3)**
>
> **3. Lack of discussion of latest works**
>
> Thank you for pointing out this problem. We agree that we should add more related work given this is a fast-moving domain. We plan to revise these paragraph for comparing most recent related works. We have added some latest related works discussions in inference acceleration and decentralized AI systems. The content of the Introduction and related work will be updated in future versions.
>
> There is the revised introduction part with the latest works:
>
> The rise of large-scale models—such as large language models (LLMs) and image generation systems—has drastically increased computational demands, prompting innovation in deployment architectures (Brown et al., 2020; Ramesh et al., 2022; Zhang et al., 2025). Traditional centralized systems, though powerful, face key limitations in scalability, data security, and cost (Li et al., 2022; Dong et al., 2025). These constraints have driven interest in decentralized architectures that distribute computation across nodes to enhance efficiency, reduce latency, and protect data privacy (Belotti et al., 2019; Zeng et al., 2024; Zhang et al., 2024).
>
> Deploying AI in distributed settings introduces distinct challenges from centralized systems, due to heterogeneous hardware, variable network latency, and dynamic load balancing requirements (Borzunov et al., 2024; Biran & Kissos, 2025; Balseiro et al., 2025). Unlike homogeneous data centers, decentralized systems span diverse devices with varying compute, memory, and communication capabilities. This heterogeneity complicates workload distribution and calls for adaptive strategies responsive to hardware availability and runtime constraints (Shen et al., 2025).
>
> **References**
>
> *   Dong, H., Jiang, J., Lu, R., Luo, J., Song, J., Li, B., Shen, Y., & Wang, Z. (2025). Beyond A Single AI Cluster: A Survey of Decentralized LLM Training. arXiv preprint arXiv:2503.11023. https://arxiv.org/abs/2503.11023
> *   Biran, Y., & Kissos, I. (2025). Adaptive Orchestration for Large-Scale Inference on Heterogeneous Accelerator Systems Balancing Cost, Performance, and Resilience. arXiv preprint arXiv:2503.20074.
> *   Balseiro, S. R., Mirrokni, V. S., & Wydrowski, B. (2025). DGD-LB: Distributed gradient descent load balancing with network latency awareness. arXiv preprint arXiv:2504.10693. https://arxiv.org/abs/2504.10693
> *   Zhang, ZX., Wen, YB., Lyu, HQ. et al. AI Computing Systems for Large Language Models Training. J. Comput. Sci. Technol. 40, 6–41 (2025). https://doi.org/10.1007/s11390-024-4178-1
> *   Shen, T., Zhu, D., Zhao, Z., Wu, C., & Wu, F. (2025). Will LLMs Scaling Hit the Wall? Breaking Barriers via Distributed Resources on Massive Edge Devices. arXiv preprint arXiv:2503.08223.
> *   Zeng, Z., Wang, J., Yang, J., Lu, Z., Zhuang, H., & Chen, C. (2024). PrivacyRestore: Privacy-Preserving Inference in Large Language Models via Privacy Removal and Restoration. arXiv preprint arXiv:2406.01394.
> *   Zhang, M., Shen, X., Cao, J., Cui, Z., & Jiang, S. (2024). Edgeshard: Efficient llm inference via collaborative edge computing. IEEE Internet of Things Journal.
>
> **4. How does the proposed MetaInf framework handle real-time adaptation to new tasks or hardware when no prior performance data is available?**
>
> MetaInf is a meta-learning framework designed to generalize across tasks and hardware configurations. While prior performance data can enhance prediction accuracy, the system is not strictly dependent on it. In scenarios where historical data is unavailable, MetaInf operates in a cold-start mode by leveraging metadata, such as model size, input length, and device specifications to make initial predictions. Furthermore, we can rapidly generate sufficient data to adapt to new tasks or hardware setups, which only involving a few experiments. In most practical deployment environments, this represents a transient challenge rather than a fundamental limitation.

---

> ### Author Response · Authors · 2025-06-03
> **Response to Reviewer bZwy (3/3)**
>
> **5. Limitations of Historical Data and Ensuring Robustness Without Prior Knowledge**
>
> Thank you for pointing out this problem. Historical data and online adaptation - this is a very good point, we did not consider temporal distribution shift over time, a straight forward fix would be an online adaptation fix. Because the meta learner training cost is very small and online adaptation is very practical in the real world setting. By using actual user data to continuously update the meta learner, MetaInf can be well adapted to new tasks and hardware.
>
> **6. Potential Failure of Meta-Learning Under Heavy Load or Resource Constraints**
>
> Thank you for pointing out this problem. Under constraint resources, if the training data does not contain very diverse data, it can result in failure cases. However, based on our experimental experience, meta learning methods can basically successfully predict the correct results when the data distribution is relatively comprehensive (hardware, model, inference parameters). Like most AI methods, we need a clean and generalized dataset as support. From generalizability experiments, we noticed that a drastic distribution shift can cause the model to fail to generalize, however this is an inherent problem to all AI models. Lastly, we also note that the capabilities of the predictor heavily relies on the knowledge and capabilities of the embedding generation model. If the produced embedding does not contain rich and up to date information, for example, fails to know the GPU or hallucinates on the description of the model architecture, it can lead to unexpected failures. These concerns can and should be controlled and fine-tuned to specific deployment environments.

---

> ### Author Response · Authors · 2025-06-09
>
> Dear Reviewers,
>
> Thank you again for the thoughtful reviews and the opportunity to improve our submission. We hope that our revisions and responses have addressed the key concerns raised during the review process.
>
> As the discussion period is nearing its end, please feel free to let us know if there are any additional questions or clarifications we can provide.
>
> If appropriate, we would also appreciate any update on the status or score of our submission. We completely understand if it is still under discussion.
>
> Best regards,
> The Authors

---

### Official Review · Reviewer_d5hq · 2025-05-13

**Rating:** 7
**Confidence:** 4
**Ethics Flag:** 1

**Summary:**

This paper highlights the challenges in choosing the correct acceleration in decentralized model deployment systems. The paper proposes to resolve these issues with a meta-learning-based framework that can automate the choice of acceleration based on the historical performance data of various acceleration techniques across different tasks

**Questions To Authors:**

1. Could you provide stronger motivation for using meta learning for acceleration selection instead of a simpler heuristic based solution given clear performance trends among Prefix Caching, Continuous Batching and Chunked Prefill across batch sizes?

2. Can you revise the writing to remove the excessive use of dashes that suggests AI generated text?

3. Can you ensure that figures are discussed in numerical order so that Figure 3 appears in the text before Figure 5?

4. Can you correct the typo in Tables 1 and 2 where “Chunked” is written as “Chucked”?

5. Can you add an explanation of the term SVD when it is first mentioned in the manuscript and add some background for it?

6. Have you considered extending the framework to multi-objective optimization, for example balancing latency against energy consumption?

**Reasons To Accept:**

1. The approach has been evaluated on multiple LLMs and datasets and also includes an ablation study.

2. The paper has considered performance acceleration tradeoffs, which is appreciated.

3. The figures are well-presented.

4. The design includes theoretical details of the proposed approach.

**Reasons To Reject:**

1. The paper does not provide sufficient motivational evidence for using meta-learning for acceleration choice; it seems that it could be resolved using a heuristics-based approach, as clear trends can be observed. For example, Prefix Caching is performing best overall, followed by Continuous Batching and Chunked Prefill for larger batch sizes, and the other way around (Chunked Prefill and Continuous Batching) for smaller batch sizes. So it would be much simpler to develop a heursitics based solution.
2. The text in the paper seems AI-generated, including too many dashes, which gives an indication of typically AI-generated text.
3. The paper is not well written. Figures are not discussed in order; Figure 5 is discussed before Figure 3 in the text.

Minor:

In Tables 1 and 2, "Chunked" is written as "Chucked".

The term "SVD" is not explained.

---

> ### Author Response · Authors · 2025-06-02
> **Response to Reviewer d5hq**
>
> Thank you for your insightful comments. We have also incorporated your suggestions into the revised manuscript and provided detailed explanations for each question below. We hope that our revisions and rebuttal could address your concerns and that you might consider raising the rating. Please let us know if you have any further questions or require additional results.
>
> **1. Motivation for using meta-learning method**
>
> | Model|Chunked Prefill|Prefix Caching|All Methods|
> | ----------------------------- | ---------------:| ----------------:| ---------------:|
> | Baichuan2-7B-Chat|+3.82%|**+37.63%**|+7.96%|
> | Qwen2.5-7B-Instruct-1M|**+4.15%**|−10.66%|−7.20%|
> | Phi-2|−0.90%|−56.79%|−11.39%|
> |Meta-Llama-3.1-8B-Instruct|**+5.40%**|−4.03%|+0.33%|
>
> Here we performed additional experiments on NVIDIA L4 GPUs, with continuous batching turned on by default with fixed batch size of 1024 so it's solely testing for the other two acceleration methods. We notice that the for different models, each specific acceleration method behaves differently based on the model's architecture, context length and parameter size. For example, Phi-2 performs substantially worse with prefix-caching turned on while other models like Baichuan2 performs significantly better. This demonstrates the need for a predictor that understands the different attributes and can thus pick the correct method for each model and particular hardware as we proposed.
>
> **Simple Heuristic Methods Performance**
> | Method             | Acc.      | Accel.   |
> | ------------------ | --------- | -------- |
> | ISAC| 0.578| 1.10|
> | Global Best| 0.629| 1.16|
> | ARGOSMART| 0.660| 1.17|
> | ALORS| 0.725| 1.20|
> | **MetaInf (Ours)** | **0.898** | **1.55** |
>
> We have tested against heuristics and simpler methods and these methods perform measureably worse in terms of performance in our experiments. The following table demonstrates the time savings of inference providers and the amortized time savings will be significant.
>
> | Model| Time (s)|
> | ------------------| ---------|
> | Llama3.1-8B-Instruct |193.8|
> | **MetaInf(ours)** | **0.2** |
> | **LLama + MetaInf(Ours)**| **148.2**|
>
> We measured that the Llama3.1-8B-Instruct model with only continuous batching over the ShareGPT dataset with 2000 prompts and 10 max new tokens per prompt, similarly with acceleration methods chosen by our MetaInf (Chunked Prefill + Prefix Caching) and we obtain a 24% performance gain from a 0.2s inference call to determine the acceleration method to be used.
>
> **2. Revise the writing to remove the excessive use of dashes that suggests AI generated text**
> i. AI tools were used for grammar and typo check rather than fully writing the paper. The original manuscript was fully written by us. We intend to correct that in the final version, for example, the first paragraph: "The growing demand for large-scale models such as language models and image generation systems has significantly increased computational requirements. Although centralized systems remain capable, they often face challenges related to scalability, data security, and cost. These limitations have sparked interest in decentralized architectures, which distribute computation across multiple nodes to improve efficiency, reduce latency, and enhance privacy."
>
> **3. Figure number order problem**
> Due to the limited space, we moved some tables and figures into the appendix, and the choice of Figure 3 and Figure 5 is made out of formatting necessity at the time and with the revise of the rest of the manuscript, we do intend to make sure all tables and figures are discussed in sequence and correct any typos like 'Chucked' in the final manuscript.
>
> **4. The explanation of the term SVD**
> For the SVD (Singular Value Decomposition), it is our oversight to not include the full explanation and background for the term. We intend to correct this with the following adjustment to the manuscript:
> While one-hot encodings are considered in ablation, we primarily use LLM-derived embeddings that captures rich semantic information about model configurations and task metadata. To make these embeddings more tractable for lightweight meta-learners such as MetaInf, we apply Truncated Singular Value Decomposition (SVD). By projecting high-dimensional LLM embeddings onto a lower-dimensional subspace, truncated SVD reduces computational complexity and mitigates overfitting while preserving the structure and semantic variance critical for generalization.
>
> **5. Multi-objective optimization considerations**
> We appreciate the reviewer’s suggestion to explore multi-objective optimization, particularly in balancing latency with energy consumption. While our current work focuses solely on latency, we agree this is an important extension, especially in resource-constrained or edge settings. The current architecture is amenable to incorporating such objectives, e.g., by augmenting metadata and learning trade-off-aware selection policies. We view this as a promising direction for future work.

---

> > ### Author Response · Authors · 2025-06-09
> >
> > Dear Reviewers,
> >
> > Thank you again for the thoughtful reviews and the opportunity to improve our submission. We hope that our revisions and responses have addressed the key concerns raised during the review process.
> >
> > As the discussion period is nearing its end, please feel free to let us know if there are any additional questions or clarifications we can provide.
> >
> > If appropriate, we would also appreciate any update on the status or score of our submission. We completely understand if it is still under discussion.
> >
> > Best regards,
> > The Authors

---

> ### Comment · Reviewer_d5hq · 2025-06-10
>
> I want to thank the authors for providing an explanation of my concerns and appreciate their efforts! The new results resolve my concerns about motivation, I highly recommend that these results be added to the motivation of the paper, which currently is lacking.  I also highly encourage the authors to address the numerous writing issues present in the current manuscript.
>
> Based on the explanation and the new results, while appreciating the efforts put in by the authors in the rebuttal phase, I have increased my score to 7.

---

### Author Response · Authors · 2025-06-02
**General Response and Contributions**

We sincerely thank all reviewers for their thoughtful and constructive feedback. The insights provided have been invaluable in improving both the clarity and rigor of our work. We have carefully addressed each comment individually and summarize the key revisions and clarifications made in response to your suggestions below.

In addition to responding to specific reviewers, in this response, we would like to highlight our empirical novelty and contributions to the community.

**Contributions**

Our paper proposes MetaInf, a novel framework tackling the critical open question of how to dynamically select optimal inference acceleration strategies for Large Language Models (LLMs) in complex, heterogeneous, and resource-constrained decentralized environments. Prior approaches often rely on static rules, expert intuition, or costly online benchmarking for each new scenario. We empirically demonstrate that MetaInf, through meta-learning and rich semantic embeddings of system components, enables adaptive, zero-shot, and budget-aware selection of these strategies. This is a capability not systematically shown before for this specific and increasingly important challenge. We list our key contributions as follows:

*   **Systematic Empirical Validation of Adaptive Selection for LLM Inference:**
    Our primary contribution is the rigorous empirical validation of MetaInf for selecting LLM inference acceleration strategies. We comprehensively demonstrate its superiority over diverse heuristic and traditional learning-based baselines across varied LLMs, hardware platforms, and workloads. This establishes the significant benefits of an adaptive, meta-learning approach in complex decentralized environments, moving beyond one-size-fits-all solutions.

*   **Novel Integration of LLM-derived Semantic Embeddings for System State Representation:**
    We introduce and validate the use of LLM-derived semantic embeddings to represent heterogeneous system components, including dataset characteristics, model architectures, and hardware profiles. This contrasts with traditional reliance on hand-crafted numerical features or simpler encodings. Our ablation studies show these rich embeddings are key to MetaInf's ability to achieve effective zero-shot generalization to unseen configurations.

*   **A Practical Framework for Budget-Constrained, Zero-Shot Inference Optimization in Decentralized Systems:**
    MetaInf provides a practical solution by explicitly incorporating budget constraints into the acceleration strategy selection process, a critical aspect for real-world deployments. The framework performs zero-shot online selection by leveraging a pre-trained meta-predictor. This approach eliminates the reliance on costly real-time measurements for new tasks, making MetaInf highly suitable for dynamic decentralized environments. This framework offers a path towards more efficient and economically viable deployment of large AI models.

---

### Decision · Program_Chairs · 2025-07-08

**Decision:**

Accept

**Comment:**

This paper introduces a meta-learning framework to select the optimal acceleration strategies (e.g., prefix caching and continuous batching) for large AI models in the distributed settings. The proposed framework outperforms multiple baselines, and all reviewers are leaning toward accepting the submission. The authors provided additional experimental results and discussion during the rebuttal, which should be incorporated to the final version of the submission.